# Eddy Current Testing of Conductive Coatings Using a Pot-Core Sensor

**DOI:** 10.3390/s23021042

**Published:** 2023-01-16

**Authors:** Grzegorz Tytko

**Affiliations:** Faculty of Automatic Control, Electronics and Computer Science, Silesian University of Technology, Akademicka 16, 44-100 Gliwice, Poland; grzegorz.tytko@polsl.pl

**Keywords:** eddy current testing, pot-core sensor, coatings, analytical modeling, sensor impedance, truncated region eigenfunction expansion method, thermal barrier coating

## Abstract

Conductors consisting of thin layers are commonly used in many industries as protective, insulating or thermal barrier coatings (TBC). Nondestructive testing of these types of structures allows one to determine their dimensions and technical condition, while also detecting defects, which significantly reduces the risk of failures and accidents. This work presents an eddy current system for testing thin layers and coatings, which has never been presented before. It consists of an analytical model and a pot-core sensor. The analytical model was derived through the employment of the truncated region eigenfunction expansion (TREE) method. The final formulas for the sensor impedance have been presented in a closed form and implemented in Matlab. The results of the calculations of the pot-core sensor impedance for thin layers with a thickness above 0.1 mm were compared with the measurement results. The calculations made for the TBC were verified with a numerical model created using the finite element method (FEM) in Comsol Multiphysics. In all the cases, the error in determining changes in the components of the pot-core sensor impedance was less than 4%. At the same time, it was shown that the sensitivity of the applied pot-core sensor in the case of thin-layer testing is much higher than the sensitivity of the air-core sensor and the I-core sensor.

## 1. Introduction

Critical elements used in the aerospace, energy, chemical engineering or petrochemical industry are very often covered with protective coatings made of high-quality materials. Coatings of this type are applied on the external and internal surfaces of pipelines [1], on cladded conductors utilised in aerospace engineering [2], on gas turbine blades [3] and on aeroengine blades [4,5]. The multilayer structure increases wear resistance and provides protection against corrosion and oxidation, thus extending the lifetime of the material and reducing the likelihood of its failure. Coatings may also provide thermal insulation, which in some applications facilitates reduction of energy loss, whereas in others it allows the utilisation of materials in working conditions involving temperatures exceeding even 1000 °C. In the latter case, the layer structure of the thermal barrier coating (TBC) type [6,7,8] can reduce the substrate surface temperature by more than 100 °C.

The most commonly used coating process enables one to distinguish three layers in the obtained structure. Both the substrate (bottom layer) and the bond coat (middle layer) are made of electrically conductive material. The third layer is a nonconductive top coat (upper layer), which constitutes additional protection. All three layers are exposed to various types of damage and loss of their properties due to the unfavourable working environment, i.e., high temperature, stress, humidity or high pressure. The most common defects include thickness loss, material degradation, delamination, cracks and corrosion. Each of these defects affects the structure of protective coatings and poses a potential hazard of a serious accident, leakage or even catastrophe. The probability of this type of failure can be significantly reduced by periodically using nondestructive evaluation.

The eddy current technique is a nondestructive method that has been successfully used for testing metals through the measurement of the change in the impedance of a circular sensor placed near the coated surface [9]. A sensor fed with alternating current generates a magnetic field, which penetrates the top coat and induces eddy currents in the bond coat and the substrate. The eddy currents create a secondary magnetic field, which changes the sensor impedance. This change is dependent on the parameters of the sensor, the properties of the tested object and its geometry. Any loss in coating thickness, corrosion, delamination or any other defect disturbs the flow of eddy currents. As a consequence, in such cases, the impedance value is different from that of the tested component without the defect. This property also makes it possible to identify conductive objects [10,11,12] and to determine their thickness [13,14], radius [15,16], magnetic anisotropy [17] or electrical conductivity [18,19].

A comprehensive system for eddy current testing consists of a mathematical model and a sensor. The model is utilised to carry out test simulations, determine optimal test parameters and interpret the obtained measurement data [20]. The necessity to perform many thousands of iterations means that a short computation time must be the key feature of the mathematical model. The fastest models are analytical, in which—unlike in numerical models—the final formulas are presented in a closed form. However, the derivation of analytical models is complicated, and for many eddy current configurations, analytical expressions for the change in the sensor impedance have not been obtained so far.

The air-core sensor [21,22,23,24,25,26] and the I-core sensor [27,28,29,30] are utilised in eddy current testing of coatings [31,32,33,34], stratified conductors [35,36,37,38] and thermal barrier coating (TBC) [3,5,6]. The weakness of sensors of this type, however, lies in the occurrence of the leakage of some part of the magnetic flux into the air. This imperfection is not observed in the E-type pot-core sensor, which therefore ensures a much greater sensitivity. This type of sensor closes the magnetic flux inside the core and directs it straight to the surface of the tested object. Such a property makes it possible to detect even minor changes in geometric dimensions and physical properties in tested elements. However, the manufacturing of a pot-core sensor is more complicated, and an analytical model is much more difficult to obtain than in the case of the air-core sensor and the I-core sensor.

So far, the pot-core sensor has been used for testing discs [39,40] and plates [41,42,43,44] with a thickness much greater than conductive foils or typical layered structures (Figure 1). This work is the first time that an eddy current system containing a pot-core sensor and an analytical model for testing thin layers and coatings with a thickness of 0.1 mm has been used. The analytical model was derived with the employment of the truncated region eigenfunction expansion (TREE) method [45,46] and was subsequently implemented in Matlab. The results of the calculations were verified using measurements and the finite element method (FEM), and the determined error did not exceed 4%. The changes in the components of the pot-core sensor impedance were compared with the values obtained for the air-core sensor and the I-core sensor. The sensitivity of the pot-core sensor turned out to be by far the highest, which indicates promising prospects for using the proposed eddy current system for testing thin conductive layers and coatings.

## 2. Analytical Model

The analytical model of the pot-core sensor above a two-layer half-space was derived with the TREE method in [43]. In this work, using an analogous approach, this model was extended to test three-layer coatings of a finite thickness. The tested material comprised a nonconductive top coat with a thickness of *l*_1_, a bond coating with a thickness of *l*_2_ − *l*_1_, and a substrate with a thickness of *l_3_* − *l_2_*. The magnetic permeability of conductive coatings was determined as *μ*_6_, *μ*_7_, and the electrical conductivity as *σ*_6_, *σ*_7_. The problem was analysed in a cylindrical coordinate system, and the solution domain was divided into 9 regions and limited to the value of parameter *b* (Figure 2). Bounding the solution domain, i.e., limiting the range of a coordinate, results in discrete eigenvalues for that coordinate direction [47]. The discrete eigenvalues **q** of regions with a homogeneous structure (1, 5–8) are the positive real roots of the Bessel function of the first kind *J*_1_(*x*) and are calculated from equation *J*_1_(**q** *b*) = 0. Region 3 consists of 3 subregions (0 ≤ *r* ≤ *a*_1_, *a*_1_ ≤ *r* ≤ *c*_2_, and *c*_2_ ≤ *r* ≤ *b*). The discrete eigenvalues **m** of region 3 are the positive real roots of the equation $L_{1}^{\prime }(m\;b)=0$, where:(1)$$L_{1}^{\prime }(m\;r)=\frac{\pi }{2}m\;c_{2}[B_{2F}^{\prime }J_{1}(m\;r)+C_{2F}^{\prime }Y_{1}(m\;r)],$$
(2)$$C_{2F}^{\prime }=\frac{1}{\mu_{f}}J_{1}(m\;c_{2})L_{0}(m\;c_{2})-J_{0}(m\;c_{2})L_{1}(m\;c_{2}),$$
(3)$$B_{2F}^{\prime }=-\frac{1}{\mu_{f}}Y_{1}(m\;c_{2})L_{0}(m\;c_{2})+Y_{0}(m\;c_{2})L_{1}(m\;c_{2}),$$
(4)$$L_{n}(m\;r)=\frac{\pi }{2}m\;a_{1}[B_{2F}J_{n}(m\;r)+C_{2F}Y_{n}(m\;r)],$$
(5)$$C_{2F}=(\mu_{f}-1)J_{0}(m\;a_{1})J_{1}(m\;a_{1}),$$
(6)$$B_{2F}=J_{1}(m\;a_{1})Y_{0}(m\;a_{1})-\mu_{f}J_{0}(m\;a_{1})Y_{1}(m\;a_{1}).$$

The relative magnetic permeability of the pot-core sensor was determined as *μ*_f_. Regions 3 and 4 consist of 5 subregions (0 ≤ *r* ≤ *a*_1_, *a*_1_ ≤ *r* ≤ *a*_2_, *a*_2_ ≤ *r* ≤ *c*_1_, *c*_1_ ≤ *r* ≤ *c*_2_, and *c*_2_ ≤ *r* ≤ *b*). The eigenvalues **p** of these regions were determined—using the Bessel function *Y*_n_(*x*)—from the equation:
(7)$$R_{1}^{\prime \prime \prime }(p\;b)=0$$
where:(8)$$R_{n}^{\prime \prime \prime }(p\;r)=\frac{\pi }{2}p\;c_{2}[B_{3F}^{\prime \prime \prime }J_{n}(p\;r)+C_{3F}^{\prime \prime \prime }Y_{n}(p\;r)],$$
(9)$$C_{3F}^{\prime \prime \prime }=\frac{1}{\mu_{f}}J_{1}(p\;c_{2})R_{0}^{\prime \prime }(p\;c_{2})-J_{0}(p\;c_{2})R_{1}^{\prime \prime }(p\;c_{2}),$$
(10)$$B_{3F}^{\prime \prime \prime }=-\frac{1}{\mu_{f}}Y_{1}(p\;c_{2})R_{0}^{\prime \prime }(p\;c_{2})+Y_{0}(p\;c_{2})R_{1}^{\prime \prime }(p\;c_{2}),$$
(11)$$R_{n}^{\prime \prime }(p\;r)=\frac{\pi }{2}p\;c_{1}[B_{3F}^{\prime \prime }J_{n}(p\;r)+C_{3F}^{\prime \prime }Y_{n}(p\;r)],$$
(12)$$C_{3F}^{\prime \prime }=\mu_{f}J_{1}(p\;c_{1})R_{0}^{\prime }(p\;c_{1})-J_{0}(p\;c_{1})R_{1}^{\prime }(p\;c_{1}),$$
(13)$$B_{3F}^{\prime \prime }=-\mu_{f}Y_{1}(p\;c_{1})R_{0}^{\prime }(p\;c_{1})+Y_{0}(p\;c_{1})R_{1}^{\prime }(p\;c_{1}),$$
(14)$$R_{n}^{\prime }(p\;r)=\frac{\pi }{2}p\;a_{2}B_{3F}^{\prime }J_{n}(p\;r)+C_{3F}^{\prime }Y_{n}(p\;r),$$
(15)$$C_{3F}^{\prime }=\frac{1}{\mu_{f}}J_{1}(p\;a_{2})R_{0}(p\;a_{2})-J_{0}(p\;a_{2})R_{1}(p\;a_{2}),$$
(16)$$B_{3F}^{\prime }=-\frac{1}{\mu_{f}}Y_{1}(p\;a_{2})R_{0}(p\;a_{2})+Y_{0}(p\;a_{2})R_{1}(p\;a_{2}),$$
(17)$$R_{n}(p\;r)=\frac{\pi }{2}p\;a_{1}[B_{3F}J_{n}(p\;r)+C_{3F}Y_{n}(p\;r)],$$
(18)$$C_{3F}=(\mu_{f}-1)J_{0}(p\;a_{1})J_{1}(p\;a_{1}),$$
(19)$$B_{3F}=\mu_{f}J_{0}(p\;a_{1})Y_{1}(p\;a_{1})-J_{1}(p\;a_{1})Y_{0}(p\;a_{1}).$$

The calculation of the eigenvalues allows one to determine the expressions for the magnetic vector potential of the coil. For this purpose, a filamentary coil was used, all of whose turns concentrated in a circle of radius *r*_0_ were placed at a distance *h*_0_ from the three-layer conductive structure. At first, the magnetic vector potential for the filamentary coil (*r*_2_ − *r*_1_ → 0, *h*_2_ − *h*_1_ → 0) was written in the matrix notation:(20)$$A_{1}(r,z)=J_{1}(q\;r)q^{-1}e^{-qz}C_{1},$$
(21)$$A_{2}(r,z)=\begin{array}{lll} J_{1}(m\;r) & & 0\leq r\leq a_{1}\\ L_{1}(m\;r) & m^{-1}(e^{-mz}C_{2}-e^{mz}B_{2}), & a_{1}\leq r\leq c_{2}\\ L_{1}^{\prime }(m\;r) & & c_{2}\leq r\leq b \end{array}$$
(22)$$A_{3}(r,z)=\begin{array}{lll} J_{1}(p\;r) & & 0\leq r\leq a_{1}\\ R_{1}(p\;r) & & a_{1}\leq r\leq a_{2}\\ R_{1}^{\prime }(p\;r) & p^{-1}(e^{-pz}C_{3}-e^{pz}B_{3}), & a_{2}\leq r\leq c_{1}\\ R_{1}^{\prime \prime }(p\;r) & & c_{1}\leq r\leq c_{2}\\ R_{1}^{\prime \prime \prime }(p\;r) & & c_{2}\leq r\leq b \end{array}$$
(23)$$A_{4}(r,z)=\begin{array}{lll} J_{1}(p\;r) & & 0\leq r\leq a_{1}\\ R_{1}(p\;r) & & a_{1}\leq r\leq a_{2}\\ R_{1}^{\prime }(p\;r) & p^{-1}(e^{-pz}C_{4}-e^{pz}B_{4}), & a_{2}\leq r\leq c_{1}\\ R_{1}^{\prime \prime }(p\;r) & & c_{1}\leq r\leq c_{2}\\ R_{1}^{\prime \prime \prime }(p\;r) & & c_{2}\leq r\leq b \end{array}$$
(24)$$A_{5}(r,z)=J_{1}(q\;r)q^{-1}(e^{-qz}C_{5}-e^{qz}B_{5}),$$
(25)$$A_{6}(r,z)=J_{1}(q\;r)s_{6}^{-1}(e^{-s_{6}z}C_{6}-e^{s_{6}z}B_{6}),$$
(26)$$A_{7}(r,z)=J_{1}(q\;r)s_{7}^{-1}(e^{-s_{7}z}C_{7}-e^{s_{7}z}B_{7}),$$
(27)$$A_{8}(r,z)=-J_{1}(q\;r)q^{-1}e^{qz}B_{8},$$
where **s**_i_ = (**q**^2^ + *j ω μ*_i_ *μ*_0_ *σ*_i_)^1/2^, and **B**_i_, **C**_i_ are the unknown coefficients.

Using the magnetic field continuity conditions for the adjacent regions, a system of 14 interface equations was created. Finding the solution of the system made it possible to determine the coefficients **B**_i_, **C**_i_:(28)$$\begin{array}{c} B_{28}\\ C_{28} \end{array}=e^{\mp md_{1}}F^{-1}[(H\pm G)e^{pd_{1}}B_{48}+(H\mp G)e^{-pd_{1}}C_{48}],$$
(29)$$\begin{array}{c} B_{48}\\ C_{48} \end{array}=D^{-1}[(H^{\prime }\pm G^{\prime })B_{58}+(H^{\prime }\mp G^{\prime })C_{58}],$$
(30)$$\begin{array}{c} B_{58}\\ C_{58} \end{array}=e^{\pm ql_{1}}[(\frac{1}{\mu_{6}}\pm qs_{6}{}^{-1})e^{-s_{6}l_{1}}B_{68}+(\frac{1}{\mu_{6}}\mp qs_{6}{}^{-1})e^{s_{6}l_{1}}C_{68}],$$
(31)$$\begin{array}{c} B_{68}\\ C_{68} \end{array}=e^{\pm s_{6}l_{2}}[(\frac{\mu_{6}}{\mu_{7}}\pm s_{6}s_{7}^{-1})e^{-s_{7}l_{2}}B_{78}+(\frac{\mu_{6}}{\mu_{7}}\mp s_{6}s_{7}^{-1})e^{s_{7}l_{2}}C_{78}],$$
(32)$$\begin{array}{c} B_{78}\\ C_{78} \end{array}=e^{\pm s_{7}l_{3}}(\mu_{7}\pm s_{7}q^{-1}),$$
where **B**_i8_ = **B**_i_/**B**_8_, **C**_i8_ = **C**_i_/**B**_8_ and **F**, **H**, **G**, **D**, **H′**, **G′** are matrices defined in the Appendix A.

The determination of the **B**_i_, **C**_i_ coefficients enables the calculation of the pot-core sensor impedance according to the formula:
(33)$$\begin{array}{l} Z=\frac{j\omega \pi \mu_{0}N^{2}}{{\left[(h_{2}-h_{1})(r_{2}-r_{1})\right]}^{2}}\chi (p^{T})p^{-1}\{[2p(h_{2}-h_{1})-e^{ph_{2}}e^{-ph_{1}}+e^{-ph_{2}}e^{ph_{1}}]+\\ {}[(e^{-ph_{1}}-e^{-ph_{2}})C_{49}-(e^{ph_{2}}-e^{ph_{1}})B_{49}]\cdot (\frac{T-U}{T+U}e^{-2md_{2}}C_{29}-B_{29})-1\\ \cdot (\lambda_{1}-\frac{T-U}{T+U}e^{-2md_{2}}\lambda_{2})\}D^{-1}\chi (p), \end{array}$$
where:(34)$$\chi (x)=\int_{xr_{1}}^{xr_{2}}rR_{1}^{\prime }(xr)dr,$$
(35)$$\lambda_{1}=F^{-1}[(H-G)e^{-pd_{1}}(e^{ph_{2}}-e^{ph_{1}})+(H+G)e^{pd_{1}}(e^{-ph_{1}}-e^{-ph_{2}})]e^{-md_{1}},$$
(36)$$\lambda_{2}=F^{-1}[(H+G)e^{-pd_{1}}(e^{ph_{2}}-e^{ph_{1}})+(H-G)e^{pd_{1}}(e^{-ph_{1}}-e^{-ph_{2}})]e^{md_{1}},$$
where **T**, **U** are matrices defined in the Appendix A.

## 3. Results

The analytical model was implemented in Matlab, and the final Formula (33) was used to calculate the sensor impedance. The measurements were carried out using the sensors and material samples shown in Figure 3. The pot-core sensor was placed in the head, facilitating the measurements, and the I-core sensor was made in a configuration with a removable core. The geometric dimensions and parameters of the sensors are shown in Table 1. The measurements of the impedance components were carried out with the Agilent E4980A precision LCR meter. In the first step, the impedance of the sensor *Z*_0_ = *R*_0_ + *jωX*_0_ in the space without conductive material was determined. In the case of the mathematical model, it was assumed that the bond coat and the substrate were nonconductive, i.e., *σ*_6_ = *σ*_7_ = 0. Then, the impedance of the sensor *Z* = *R* + *jωX* was determined after having placed it on the surface of the tested sample. All impedance measurements were performed three times, and subsequently their arithmetic mean was calculated. The values of the changes in the sensor impedance were presented as Δ*Z* = *Z* − *Z*_0_.

The first test sample consisted of a 0.5-mm-thick aluminium layer with a conductivity of 36.26 MS/m, on whose surface a 0.2-mm-thick copper layer with a conductivity of 58.38 MS/m was placed. The top layer was a 0.15-mm-thick pad made of nonconductive material. The other sample was much thinner than the first one. The same nonconductive pad was used, under which a layer of copper with a thickness of 0.1 mm and a conductivity of 58.49 MS/m was placed. The substrate was made of brass with a thickness of 0.28 mm and a conductivity of 17.25 MS/m. The values of the changes in resistance Δ*R* = *R* − *R*_0_ and reactance Δ*X* = *X* − *X*_0_ obtained through the testing of both samples were normalised to reactance *X*_0_ and are presented in Figure 4, Figure 5, Figure 6 and Figure 7. The measurements were made for 40 frequency values within a range of 1 kHz to 50 kHz.

In the next step, the possibility of using the proposed analytical model for thermal barrier coating tests was examined. For this purpose, coatings with parameters corresponding to TBC utilised to protect turbine blades were modelled. The substrate had a thickness of 2 mm and a conductivity of 0.5 MS/m, and the bond coat was 0.1 mm and 0.15 MS/m, respectively. The 0.2-mm-thick top coating was nonconductive. The finite element method was used to verify the obtained results. The numerical model created in Comsol Multiphysics consisted of 31,540 triangular elements, 16,982 mesh vertices, 5925 boundary elements and 30 vertex elements. The normalised values of changes in the impedance components of the pot-core sensor for frequencies ranging from 1 kHz to 1 MHz are shown in Figure 8.

## 4. Discussion

The normalised changes in the components of the sensor impedance (Figure 4, Figure 5, Figure 6 and Figure 7) show that the sensitivity of the pot-core sensor is much higher than that of the air-core sensor and the I-core sensor. In the case of normalised changes in resistance, the difference between the sensors decreases together with the increase of frequency because the depth of eddy currents’ penetration decreases. In the initial frequency range, it may be observed that the change in the normalised reactance of the I-core sensor is bigger than that of the pot-core sensor. This difference is slight and occurs at low frequencies. Nevertheless, in the case of reactance, what is crucial is a high-frequency range, where the change in reactance is the biggest.

The frequency of the current that supplies the sensor is one of the most important parameters for performing eddy current testing, since the correctly selected frequency makes it possible to obtain the appropriate depth of penetration and sensitivity of the sensor. The largest normalised changes in resistance were obtained for the frequencies *f* = 2 kHz (sample 1) and *f* = 5 kHz (sample 2). These are the optimal frequency values for performing tests that are most affected by the electrical conductivity of the test sample. Low conductivity values are often found in layers used in thermal barrier coatings (below 1 MS/m). It is for this reason that the largest change in the resistance of the pot-core sensor in the calculations for TBC (Figure 8) was obtained for a frequency of 65 kHz. Such a large diversity points to the fact that the most advantageous approach in eddy current testing is to apply different frequency values, selected in the following way:-relatively low frequency in order to obtain the highest sensitivity of the sensor resistance,-high frequency in order to ensure the sensitivity of the imaginary part of the sensor impedance that is much higher than that of the real part.

The results of the calculations carried out using the analytical model showed good agreement in comparison to the measurements and results from the FEM numerical model. In the entire frequency range, the error in determining changes in the sensor impedance components was less than 4%. It took about 0.5 s for the TREE model to perform a single iteration consisting in the determination of the change in sensor impedance. The short calculation time and high accuracy make the analytical model of the pot-core sensor suitable for eddy current testing of thin layers, both for low and high frequencies (i.e., 1 kHz–1 MHz).

## 5. Conclusions

The eddy current system proposed in this paper, consisting of an analytical model and a pot-core sensor, was successfully adapted to testing thin layers and coatings. The measurements were made for two three-layer samples using sensors with an air-core, an I-core and a pot-core. The calculations were also carried out for TBC coatings in a frequency range of 1 kHz to 1 MHz, which were verified with the FEM model. An acceptable error of less than 4% was obtained in all cases. The change in the sensor impedance was the tested parameter, and the obtained results point to the following conclusions:

In the case of testing thin conductive layers, the pot-core sensor has a much greater sensitivity than both the I-core sensor and the air-core sensor. This greater sensitivity of the sensor makes it possible to examine thinner layers and to detect even slight disturbances in the structure of coatings. Thus, the use of the pot-core sensor in testing these types of structures should improve their current effectiveness.

The analytical model derived with the employment of the TREE method enables one to obtain accurate calculation results for thin layers made of various conductive materials. Thanks to this, it is possible to use the model to perform test simulations, as well as to interpret measurement data. Simulations allow for the determination of the expected value of the sensor impedance. When the measurement has been carried out appropriately but the result displays a deviation from the expected value, this points to the presence of a defect. Such an element or part of the structure should be replaced or subjected to more detailed testing. Due to the short calculation time, the TREE model can also be used to design a pot-core sensor with dimensions that are optimised for the purpose of testing thin layers.

The presented research will be continued, and further studies will include the testing of thin layers that contain various types of flaws, such as cracks or corrosion. We also plan to take into account the porosity and roughness of the surface and to eliminate their influence on the final result. In addition, simulations will be performed to determine the possibility of using eddy current solutions to detect very thin degradations of TBC coatings with a thickness of 10 μm.

## Figures and Tables

**Figure 1 sensors-23-01042-f001:**
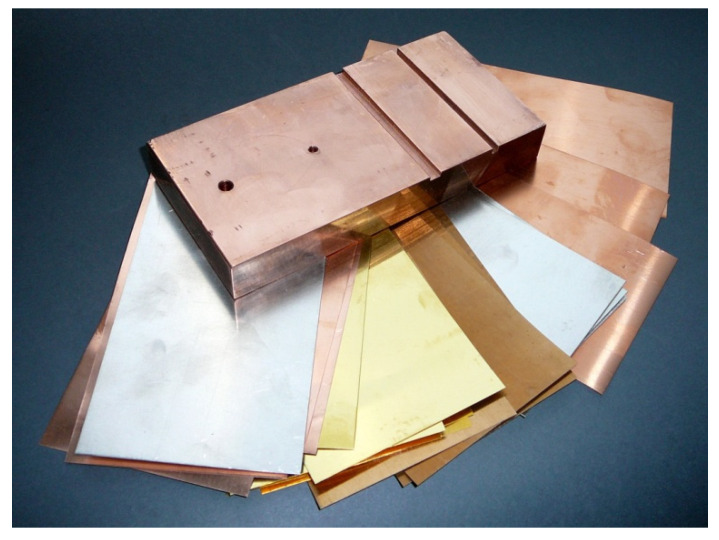
Samples for eddy current testing in the form of a thick plate made of copper (thickness 20 mm) and thin foils made of various types of conductive materials (thicknesses ranging from 0.1 mm to 0.5 mm).

**Figure 2 sensors-23-01042-f002:**
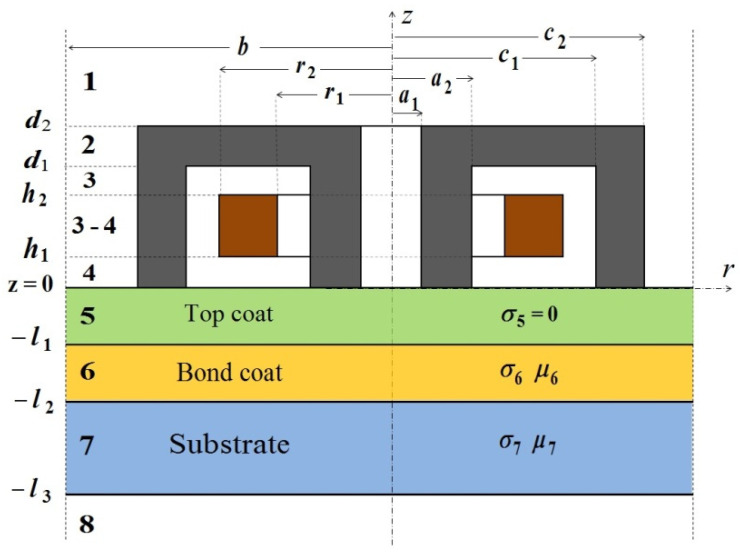
Rectangular cross-section of the pot-core sensor placed above a three-layer conductive structure.

**Figure 3 sensors-23-01042-f003:**
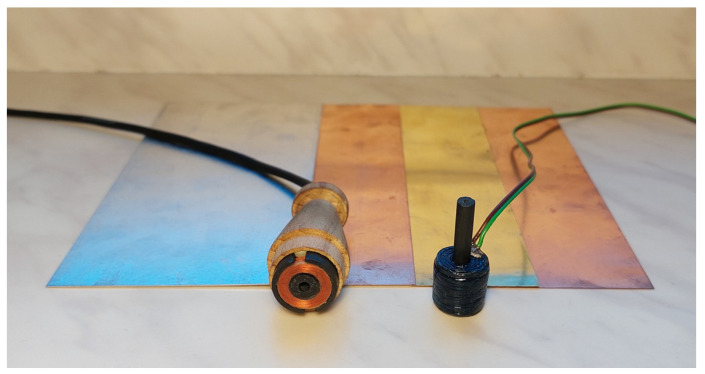
E-core sensor, I-core sensor and samples made of aluminium (*σ* = 36.26 MS/m), copper (*σ* = 58.38 MS/m), brass (*σ* = 17.25 MS/m) and copper (*σ* = 58.49 MS/m), used in the tests.

**Figure 4 sensors-23-01042-f004:**
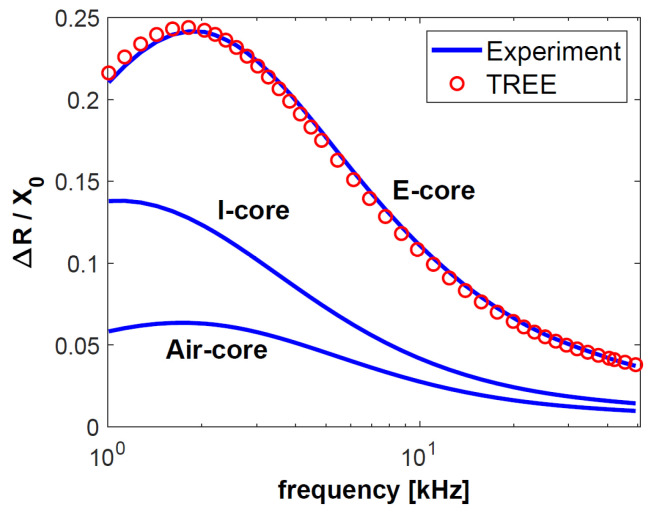
The changes in the resistance Δ*R* normalised with respect to the reactance *X*_0_ for sample made of aluminium and copper.

**Figure 5 sensors-23-01042-f005:**
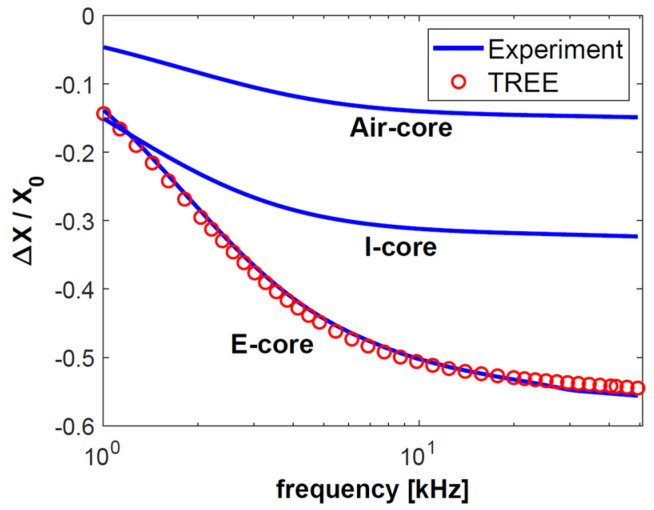
The changes in the reactance Δ*X* normalised with respect to the reactance *X*_0_ for sample made of aluminium and copper.

**Figure 6 sensors-23-01042-f006:**
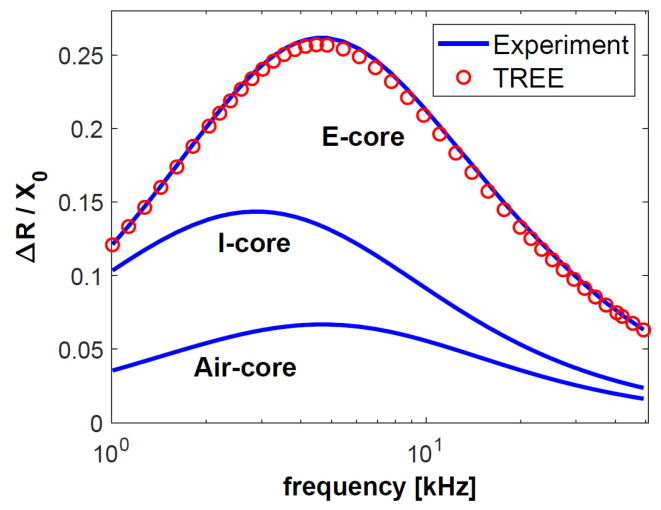
The changes in the resistance Δ*R* normalised with respect to the reactance *X*_0_ for sample made of brass and copper.

**Figure 7 sensors-23-01042-f007:**
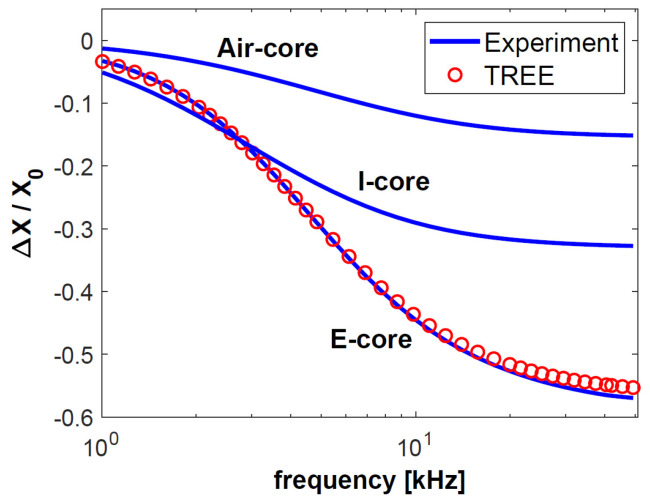
The changes in the reactance Δ*X* normalised with respect to the reactance *X*_0_ for sample made of brass and copper.

**Figure 8 sensors-23-01042-f008:**
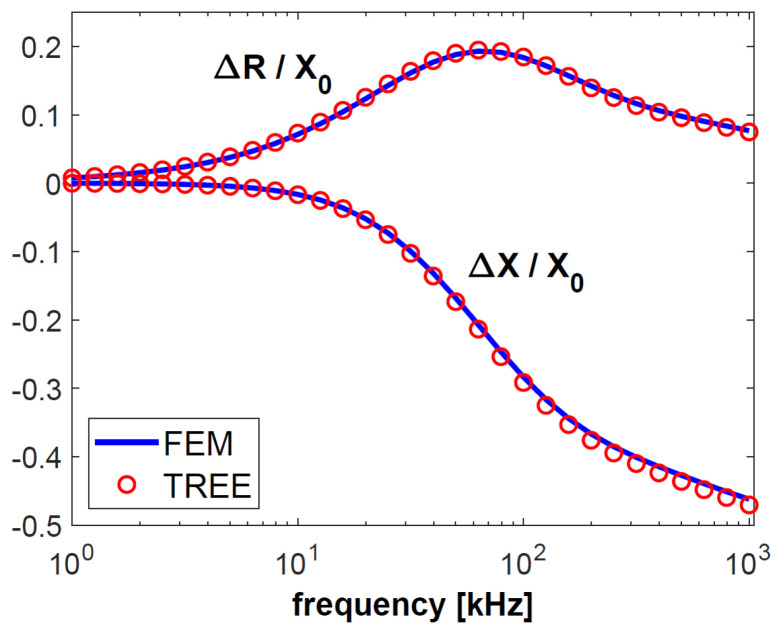
Normalised changes in the sensor resistance Δ*R* and reactance Δ*X* for thermal barrier coatings.

**Table 1 sensors-23-01042-t001:** Geometric dimensions and parameters of the sensors.

	I-Core Sensor	E-Core Sensor
Number of turns *N*	480	646
Inner coil radius *r*_1_	2.6 mm	4.3 mm
Outer coil radius *r*_2_	7.8 mm	7.3 mm
Parameter *h*_1_	0.1 mm	0.2 mm
Parameter *h*_2_	15.9 mm	3.6 mm
Inner column radius *a*_1_	0.7 mm	1.5 mm
Outer column radius *a*_2_	2.5 mm	3.7 mm
Inner core radius *c*_1_	-	7.7 mm
Outer core radius *c*_2_	-	9.1 mm
Inner core height *d*_1_	-	3.7 mm
Outer core height *d*_2_	34.5 mm	5.3 mm

## Data Availability

Not applicable.

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
