# Peer review of "Eddy Current Testing of Conductive Coatings Using a Pot-Core Sensor"

_sensors, 2023, doi:10.3390/s23021042_

Round 1
Reviewer 1 Report
This paper presents an interesting development of the TREE method for the treatment of sensors equipped with ferrite core. I think that the seminal work of Theodoulidis and Kriezis (inventors of the methode name, after all) should be cited:
- Theodoulidis TP, Kriezis EE. 2006 Eddy current canonical problems (with applications to nondestructive evaluation). Forsyth, GA: Tech Science Press
The theoretical development are difficult to follow, as they refer a lot to previous works in a not so clear way. I think that a reader with no prior knowledge about the method would be completely lost. This should be revised and presented in a more compact way: why not only develop the final expressions in a way that readers can reproduce the results presented?
As a minor remark: The quantities T and U used in the development are not defined nor referred to.
Reviewer 2 Report
sensors-2122452
Eddy current testing of conductive coatings using a pot-core sensor
The manuscript must include some essential sections, e.g., analytical studies, simulation results etc.
Some
points need to be known.
· Page 1 abstract. Author claim that "the analytical model was derived with the employment of the truncated region eigenfunction expansion (TREE) method." It will be good to add a separate section for the Analytical model and explain it in detail.
· Page 1 abstract. "The calculations made for the TBC were verified with a numerical model created using the finite element method (FEM) in Comsol Multiphysics." It will be good to add a separate section for the finite element method (FEM) in Comsol Multiphysics, showing some simulated results.
· Figure 1 and Figure 3 should explain more clearly and mention the part names in the figures.
· The author should include a brief overview of the air–core, pot-core, and I-core sensors.
· Is there any specific reason to select the frequency range from 1 kHz to 50 kHz? What about higher frequencies, i.e., in MHz etc.?
Round 2
Reviewer 1 Report
I would like to thank the authors for the modifications of the paper, which is now in my opinion much improved. I have no further remark.
Reviewer 2 Report
Sensors- 2122452
Eddy current testing of conductive coatings using a pot-core sensor
Thank you for allowing me to revise resubmitted manuscript titled " Eddy current testing of conductive coatings using a pot-core sensor" I believe the submitted manuscript and presented work are suitable for publishing in the Sensors. I suggest accepting the manuscript.